# A Psychological Ownership Based Design Tool to Close the Resource Loop in Product Service Systems: A Bike Sharing Case

**Dirk Ploos van Amstel [1,2,\*], Lenneke Kuijer [2], Remko van der Lugt [1] and Berry Eggen [2]**

[1] Research Group Co-Design, Research Centre for Learning and Innovation, HU University of Applied Sciences, Padualaan 97, 3584 CH Utrecht, The Netherlands; remko.vanderlugt@hu.nl

[2] Department of Industrial Design, Eindhoven University of Technology, Groeneloper 3, 5612 AZ Eindhoven, The Netherlands; s.c.kuijer@tue.nl (L.K.); j.h.eggen@tue.nl (B.E.)

[\*] Correspondence: dirk.ploosvanamstel@hu.nl

**Abstract:** Closing the loop of products and materials in Product Service Systems (PSS) can be approached by designers in several ways. One promising strategy is to invoke a greater sense of ownership of the products and materials that are used within a PSS. To develop and evaluate a design tool in the context of PSS, our case study focused on a bicycle sharing service. The central question was whether and how designers can be supported with a design tool, based on psychological ownership, to involve users in closing the loop activities. We developed a PSS design tool based on psychological ownership literature and implemented it in a range of design iterations. This resulted in ten design proposals and two implemented design interventions. To evaluate the design tool, 42 project members were interviewed about their design process. The design interventions were evaluated through site visits, an interview with the bicycle repairer responsible, and nine users of the bicycle service. We conclude that a psychological ownership-based design tool shows potential to contribute to closing the resource loop by allowing end users and service provider of PSS to collaborate on repair and maintenance activities. Our evaluation resulted in suggestions for revising the psychological ownership design tool, including adding 'Giving Feedback' to the list of affordances, prioritizing 'Enabling' and 'Simplification' over others and recognize a reciprocal relationship between service provider and service user when closing the loop activities.

**Keywords:** psychological ownership; product service systems; design performance; repair & maintenance; user involvement

## 1. Introduction

Products are increasingly available for use without ownership, a phenomenon also known as Product Service Systems (PSSs) [1,2]. The bicycle service of The Student Hotel (TSH)—a Dutch concept that offers international students furnished housing and complementary services is a recent example of such a PSS. It is a designed mix of tangible products (a bicycle, a rental application) and intangible services (use, storage, repair and maintenance), aimed at achieving the consumer's goals (getting from A to B). Products within a PSS are used without personal ownership. This can have a negative effect on the environmental-impact of products, as a lack of a sense of ownership can potentially lead to a less caring handling [3]. As Peck et al. [4] put it: "The key insight of the tragedy of the commons is that shared ownership leads to a diffusion of responsibility among community members, such that no one individual steps forward to provide stewardship for the resource". Therefore, one way to make users behave more sustainably (leading to less environmental impact) compared to non-owned products is to give users a sense of ownership. This is referred to as psychological ownership [5]. Examples show that users who experience psychological ownership show signs of caring, protection, stewardship, assumption of responsibility [6] and loyalty [7]. These examples show that psychological

ownership potentially can contribute to the involvement of end users in closing the loop activities, such as repair and maintenance of bicycles in a bicycle service. For instance, Makatsoris et al. [8] found that users are willing to repair a product when they have an emotional connection with it and when they are familiar with the product (and its inner workings). Pierce, Kostova, & Dirks [4,6] propose three routes (or processes) that lead to the generation of feelings of ownership. These routes are: perceived control over; intimate knowmedge of; and self-investment into an object [9]. Research [4–6,10,11] has shown that psychological ownership can lead to more responsible handling of objects or products by end users. Peck et al. [4] show, for example, that by allowing park visitors to invest in planning their own route, these visitors are inclined to donate more (financially) to that park after the visit than people who receive a planned route. Despite these examples, little is known about how to design for psychological ownership in the context of PSS [12].

In order to arrive at well-founded design solutions in a complex context such as PSS, it can be useful for parties that deal with this context on a daily basis, such as service design agencies and PSS providers, to be guided from both theory and practical experience. Design researchers can play an important role in providing this guidance by providing their theoretical knowledge and practical experience in a methodical manner, for example by offering design methods [13,14]. This type of knowledge, generated within the design research disciplines, is referred to as 'intermediate knowledge' [15–17]. To generate this knowledge, design researchers work towards substantiated solutions by navigating back and forth between theory and practice [18] as displayed in Figure 1. On the one hand, more abstract knowledge, such as social theories and frameworks can help design researchers to arrive at "Top-Down" theoretically substantiated design solutions. On the other hand, concrete knowledge regarding the implementation of design practices can help to generate relevant and generalizable "Bottom-Up" solutions, matching the perception of all partners [17,18]. The design research process can thus be seen as a connecting process between theory and practice, which iteratively develops intermediate forms of knowledge, such as design tools, that are generalizable beyond the specific project [15].

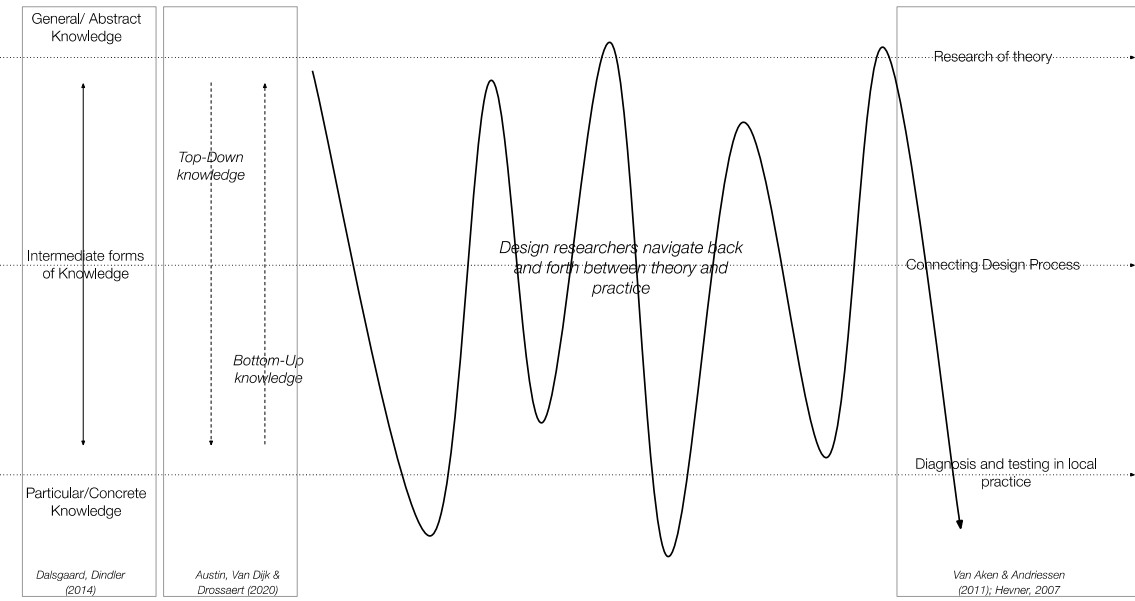

**Figure 1.** Design researchers navigate back and forth between theory and practice according to Van Aken & Andriessen [15], Hevner [16], Dalsgaard & Dindler [17] and Austin et al. [18].

This paper contributes to the described knowledge gaps of gaining intermediate knowledge while designing for psychological ownership in the context of PSS by presenting a case study in which a theoretical model of psychological ownership is applied Top-Down and the resulting design solutions are evaluated Bottom-Up in the practice of The Student

Hotel (TSH) bicycle service. In this way, a first step is taken to arrive at new intermediate forms of knowledge about designing for psychological ownership in the context of PSS and involve users in closing the loop activities. As such, our main research question was: whether and how can designers be supported with a design tool, based on psychological ownership, to involve users in closing the loop activities? To answer this question, we took a case study approach where the theoretical framework of psychological ownership was the starting point.

*The Psychological Ownership Affordances Framework*

Extending a product's lifespan can be approached from different perspectives and different approaches, such as a systemic perspective [19,20], a social practices perspective [21–23] and a Life Cycle perspective [24,25]. Within this set, we chose to focus on the approach of psychological ownership. Research shows that people exhibit higher levels of product care when it comes to things that involve perceived ownership and inciting psychological ownership seems to be a successful mechanism for triggering people to more sustainable behavior [5,9]. For example, a study by Kamleitner & Rabinovich [26] showed that sole owners exert more care towards consumer goods than joint owners. Within this body of work we selected an existing framework of psychological ownership Affordance (P.O.) [27] as starting point for our study. This framework was chosen because, to our knowledge, it is the only framework that attempts to translate knowledge about psychological ownership into design practice. The P.O. framework is based on a hierarchy of three types of goals, as defined by Hassenzahl [28]. Be-goals (on the left in Figure 2) are about the motives that people have to act, such as efficiency, effectiveness and having, finding or confirming their own identity. Motor goals are about actually performing actions, such as pedaling and steering a bicycle. Do-goals (on the middle in the model) are about the ways (routes) that are offered to us to act on these motives, for example a bicycle, which allows us to get from A to B in the city effectively. The different parts of an object have specific affordances; they tell us how to interact with it [29]. For example, think of the affordance of pedals on a bicycle, you can only do two things with it: turn it forward or backwards. By doing this, we have control over the bike. Affordances, a concept by Gibson [30] introduced in interaction design by Norman [31], can be seen as intermediate form of knowledge, which makes it possible to incorporate theoretical knowledge about psychological ownership into the design of an artifact.

Based on P.O. affordances, the sense of ownership can be increased in three ways. First, by giving users more control over products and services (Control route). Baxter et al. [32] identified five affordances that can convey perceptions of control and thus instill a sense of ownership: spatial, configuration, temporal, rate and transformation [9]. For instance, spatial control relates to the possibility to physically manipulate an object. In a country like the Netherlands, you can take and park a bike almost everywhere which contributes to the sense of ownership that is experienced. Second, the sense of ownership can be increased by having users invest in the product or service (Self-investment route). As Kamleitner & Mitchell [9] state, investment into an object can take many shapes from meaningful monetary investment to psychological effort and investment in time. Baxter et al. [27] identified five affordances that can convey perceptions of self-investment and thus instill a sense of ownership: creation, repair & maintenance, repository, emblems and preference recall. For instance, repair and maintenance activities, like inflating a bicycle tire in time, are seen as a self-investment that can increase feelings of ownership. Third, the sense of ownership can be increased by gaining intimate, personal knowledge about a product (Intimate-knowledge route). As Kamleitner & Mitchell [9] (p. 100) state, an object: "can be one's own if one feels familiar, associated with, or intimately knowledgeable about it". The six intimate-knowledge affordances as proposed by Baxter et al. [32], ageing, disclosure, periodic signaling, enabling, simplification and proximity directly relate to a user interacting with and getting to know the characteristics of an object. It is by this user-object (such as a bike) interaction in time that (stronger) feelings of ownership can grow.

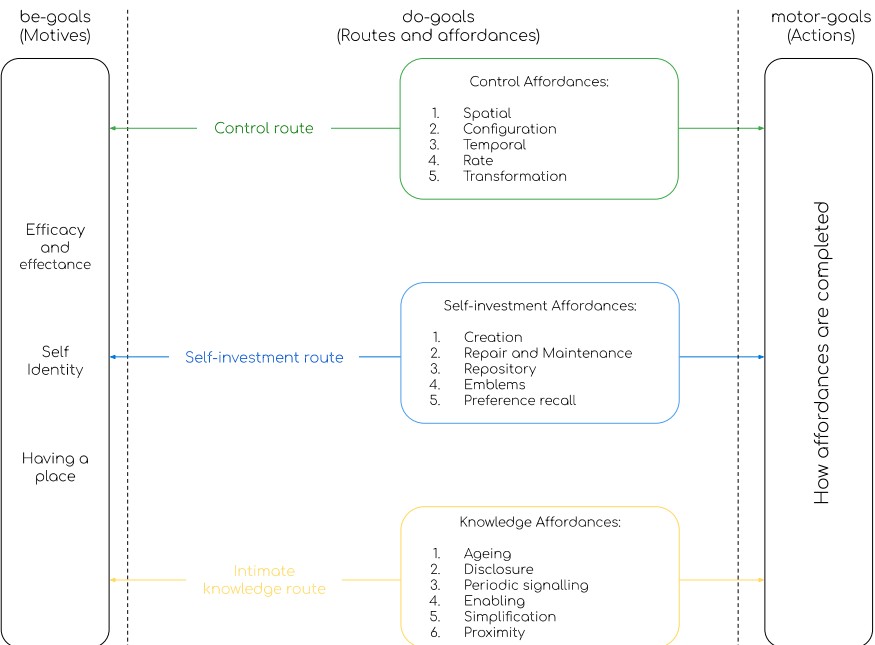

**Figure 2.** The P.O. Model Adapted with permission from Baxter & Aurisicchio [12]. 2018, Baxter & Aurisicchio.

All routes are bi-directional or, as Baxter et al. [32] argue: "Users are constantly changing through the possession process as they receive feedback from the interaction (e.g., how well objects fulfill the routes)". Despite the fact that the framework of Baxter et al. [12] makes a first translation of knowledge about psychological ownership into design practice, the textual, abstract model still needed adjustments to be applied as a tool in design practice.

## 2. Materials and Methods

### 2.1. From a Framework towards a Tool

One of the most important conditions for using a design tool as a method is that the user has confidence in the tool. As Daalhuizen [14] puts it: "Trust in a method reflects both confidence in one's ability to use the method in a way that yields desirable results as well as confidence in the applicability of the method itself to a certain goal-domain". We do not believe that simply offering a tool in a research paper textually gives this confidence to the users. In order to maximize this trust among users, it was decided to make adjustments and additions that translate the original framework to a design tool to make it more accessible and applicable for designers (as shown in Figure 2). First, we placed all sixteen affordances in the framework for an instant overview. Second, we have colored the three different routes: green for the Control route, blue for the Self-investment route and yellow for the Intimate Knowledge route. Third, we supplemented the framework with a physical and digital card set, in which all affordances per card are explained and described using one recognizable example (as shown in Figure 3a–c). Fourth, we made a presentation with an extensive explanation of the tool with examples from practice. The framework and card set were jointly available as a physical and online tool (From here, both the model and card set will be referred to as "the tools". The full version of the tool can be found online at https://www.hu.nl/onderzoek/projecten/ontwerpen-voor-duurzaam-gebruk-van-producten-binnen-product-dienst-systemen (accessed on 1 May 2022)) (See Supplementary Materials).

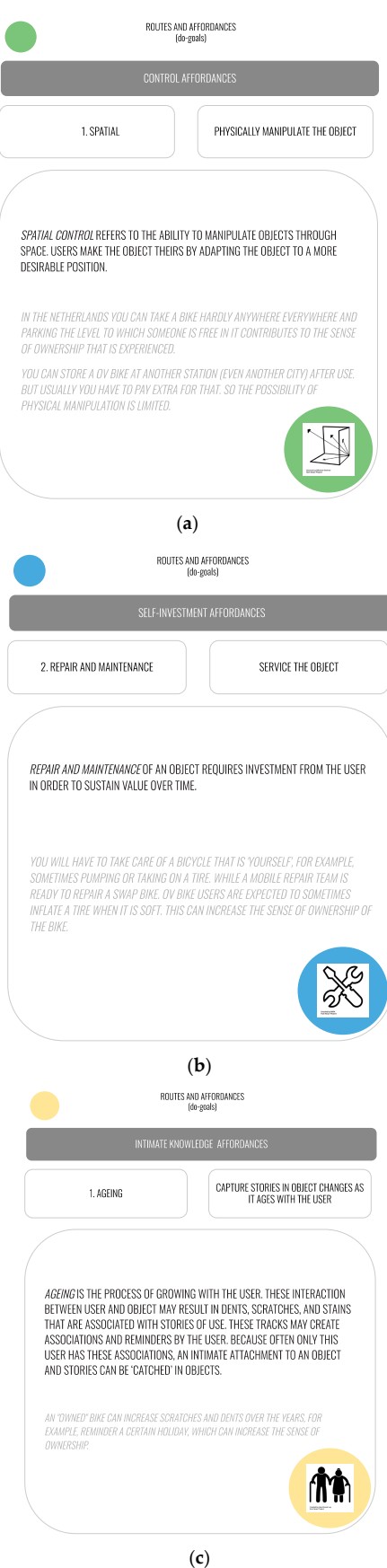

**Figure 3.** (**a**) An example of a Control Affordance Card. (**b**) An example of a Self-Investment Affordance Card. (**c**) An example of an Intimate Knowledge Affordance Card.

To gain new insights about designing for psychological ownership in a PSS, we started a case study. The study involved TSH, her partners (app supplier X-bike and circular bicycle manufacturer Roetz) and students Industrial Design at Eindhoven University of Technology (TU/e) and Communication & Multimedia Design at Utrecht University of Applied Sciences (HU) (From here on, reference will be made to "groups" and "partners"). The study was conducted between September 2019 and March 2021, as displayed in Figure 4.

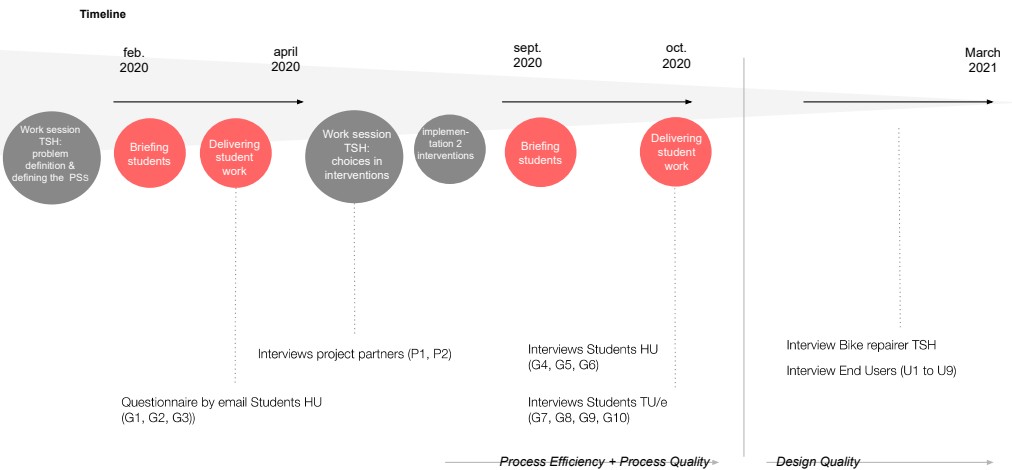

**Figure 4.** Timeline of the Case Study.

## 2.2. Work Session with TSH

During a session with TSH and partners, we first explored the problem. During the meeting, we determined that the lifespan of the used bicycles within TSH bicycle service is short, resulting in high costs and customer dissatisfaction. Before the start of the project, TSH observed two ways of irresponsible behavior by end users: (1) bicycles were not returned to the correct place after use and (2) in a defective condition. This has an impact on extending the resource loop. If you cycle on defective bicycles, there is a greater chance that materials will be damaged beyond repair. This also has a negative effect on the closing of the resource loop. After all, it is more difficult to collect defective materials for repair and reuse. After we defined the problem, we defined the PSS as the bicycle, the physical space of the bicycle parking facility (a designed mix of tangible products) and the rental app (a technology-mediated environment), following Tukker & Tischner's [2] definition as described in the introduction section.

## 2.3. Design Students at Work

We integrated the tool into two courses of HU and TU/e between February 2020 and November 2020. For HU Communication & Multimedia Design this was the group design project in the third year. For the TU/e Interaction Design students the Master's course Constructive Design Research.

As part of a briefing, we presented a design brief, and presented and distributed the tools to the student design groups. A total of ten groups, three from TU Eindhoven and seven from HU worked with the tool on the TSH case. All groups were briefed in the same way and all groups were given the same design space. That is, all groups received the same assignment description, the same toolset with brief oral explanation and were given the same freedom to use the tools as desired or not. The main question that was asked during the assignment was: how can TSH's bicycle service be (re)designed, based on insights into the relationship between ownership and use of products within a service, so that it makes a maximum contribution to responsible use of products?

The ten project outcomes were used as input for a generative session with TSH and its two partners, resulting in two implemented interventions in TSH West branch, which will be explained in the results section.

### 2.4. Interviews and Data Analysis

To analyze the Design Performance of the design tools, we conducted interviews with design students, key TSH employees and partners as well as end-users of the bike service. These interviews were structured according to the Design Performance evaluation tool developed by Tromp and Hekkert [13].

A design tool can support designers in different ways. This can be done, for example, by supporting an efficient design process or by helping to make consciously substantiated design choices during that process [14]. This can also be done by providing support by arriving at the effects as initially desired, in this particular case a longer lifespan of bicycles. All these aspects contribute to the Design Performance of a design tool [13]. In the interviews we followed three aspects of Design Performance, as proposed by Tromp and Hekkert [13]. First, Process Efficiency (PE) refers to the question to what extent a design tool helps designers to arrive at design solutions efficiently. Second, Process Quality (PQ) deals with the question to what extent designers consciously apply a design tool. In the design tool that is examined during this study, the approach has been taken from an existing abstract theoretical framework to concrete elaborations. Therefore, we can say that both PE and PQ are indicative of the degree of "Top-Down" usefulness of a theoretical framework and ensuing tools, in this case that of psychological ownership. Third, Design Quality (DQ) refers to the extent to which the design solutions are effective for the intended result (a longer lifespan of bicycles) and is indicative of connecting design solutions in the context of the relevant practice. A "Bottom-Up"- analysis of these insights can enrich and deepen the existing theoretical framework and ensuing tools.

To determine how theoretical insights "Top-Down" were applied in practice, we conducted a total of twelve interviews, involving ten groups of students and two individual partners of TSH, comprising a total of 42 interviewees. These interviews all took place after the collaboration, at the end of the project. Ten interviews were with all student project groups and two interviews involved project partners Roetz-bikes and X-Bike. The interview questions were based on Process Efficiency and Process Quality and served as a semi-structured in-depth interview (The first project groups coincided exactly with the first onset of the COVID-19 pandemic. As a result, this data was not collected through interviews, but a questionnaire by email. As a result, there was no room for in-depth exploration of the questions) (as shown in the Appendix A).

Following the field design research approach [33], we believe that the possible effects of the implemented interventions can best be determined in the daily context of the interventions. The bicycle repairperson and end users have to deal with the bicycle service in a daily context. TSH's bicycle repair person is available daily at the Amsterdam West location and is responsible for minor repairs to bicycles, such as fixing tires and repairing lamps and mudguards. From this daily role, the bicycle repair person can be seen as a committed expert and expert by experience. The end users are so-called long-stayers who stay at TSH for at least one year. They can use the bicycle service on a daily basis and can therefore be regarded as experts by experience. That is why was decided to hold interviews with both the bicycle repairer and end users. Six months after the implementation of the interventions, the bicycle repairperson and nine end users of the Amsterdam West location were interviewed about Design Quality: the possible effects of the implemented interventions in relation to the perceived degree of self-investment and intimate knowledge. The interviews were based on Design Quality and served as semi-structured in-depth interviews (as shown in Appendices B and C). Open questions were asked first, followed by closed questions. The answers gave us "Bottom-Up" insights that we could translate back into the existing theoretical framework.

To arrive at saturation of answers to the main question, all interviews with design students and partners were transcribed and the data was coded. Because the existing theoretical framework of Design Performance was the starting point for this research, it was decided to code deductively with indicators [34]. The concepts Process Efficiency, Process Quality and Design Quality have been unraveled by one researcher into concrete observable indicators. The proposed indicators were submitted to a second researcher as a check and adjusted in consultation. The text fragments from the transcripts are then linked to the indicators and compared with each other. From this comparison, the research team has arrived at saturated answers to the main questions: when and how the team members applied the design tools, how these obstructed or supported the design process, if the students and partners showed shared understanding of the intended consequences of their actions and whether and to what extent design solutions are effective for the intentional result in the eyes of the bicycle repairer and end-users.

## 3. Results

### 3.1. Design Interventions

The projects resulted in ten design proposals with rationales, as gathered from reports and interviews, displayed in Appendix D. The ten project results were used as input for a generative session with TSH and its two partners, resulting in two concepts for interventions. These two interventions were based on Intimate Knowledge and Self-Investment with the aim of involving end users in closing the resource loop activities.

### 3.1.1. Intervention 1: Intimate Knowledge

In this intervention, bicycles are offered per cluster of users (as displayed in Figure 5a,b). At the Amsterdam West branch, approximately 500 users share 180 bicycles. In the new set-up, the 500 users are divided into clusters of 50 users, who have access to 18 bicycles per cluster. This arrangement offers possibilities to increase the psychological ownership of bicycle users via the intimate knowledge route. Following the model of psychological ownership, it is expected that the feeling of ownership is increased because users use one specific bicycle (with the correct saddle height, for example) more often.

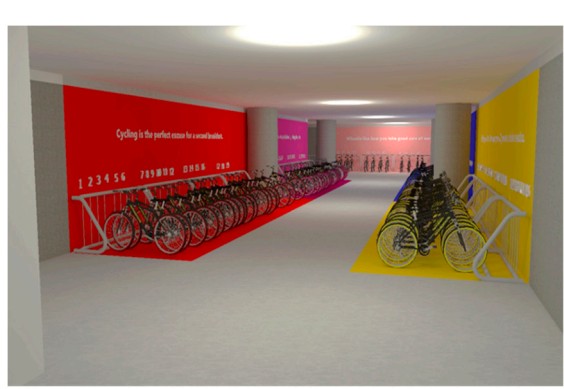
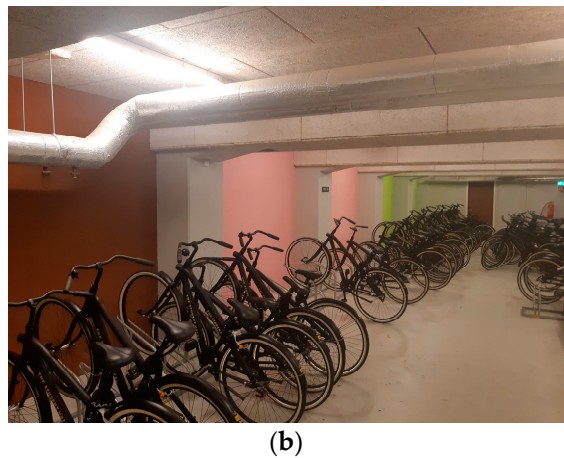

(**a**)                                                                                       (**b**)

**Figure 5.** (**a**) Clustering of bikes (Rendering). (**b**) Clustering of bikes (Real life Photo at TSH).

### 3.1.2. Intervention 2: Self-Investment

In this intervention, the user and TSH are allocated a joint responsibility for the bicycle service. The user was given *control* by having the bicycle *checked for defects* after use. When defects are observed, the user is asked via the X-Bike hire app to place them in the 'Bike Hospital' (a separated part of the bicycle storage facility, as displayed in Figure 6a,b). Because this process takes time and effort, we also allow the user to *self-invest* in the bicycle

service. The mechanics at Roetz-Bikes repair the defective bicycles, return them to the fleet and by doing so, close the resource loop.

To describe the results of our case study, the student-delivered material with rationale, the two implemented interventions and the interviews with design students, stakeholders, bicycle repairer and end-users were analyzed using the Design Performance dimensions of Process Efficiency (PE), Process Quality (PQ) and Design Quality (DQ).

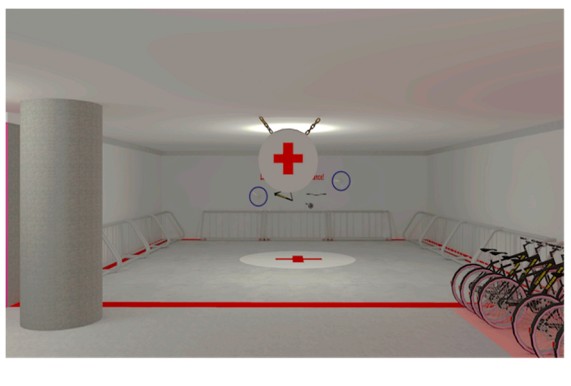

(**a**)

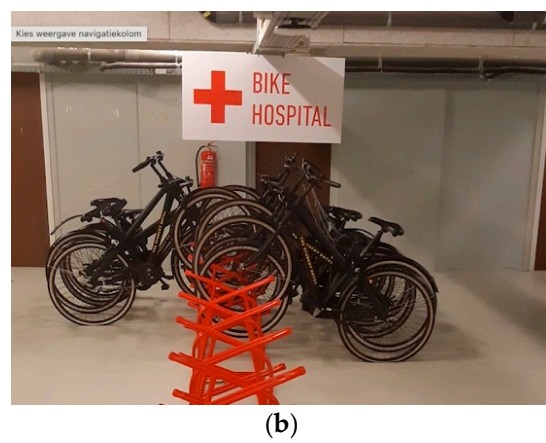

(**b**)

**Figure 6.** (**a**) Bike Hospital (Rendering). (**b**) Bike Hospital (Real Life Photo at TSH).

### 3.2. Process Efficiency

To assess the PE of the applied design tools, all design students and partners were asked about the extent to which the tools contributed to the design process. That is (1) whether the tools were used during the project and if so, (2) where in the design process, (3) what specific design choices were made using the tools and (4) whether and how the tools contributed to an efficient design process.

Eight out of ten groups indicated that they actually applied the tools during the project. The two groups that indicated that they had not used the tools said that this was because they were too complex and extensive. In their view, this, together with the short time frame of the project, created too high a barrier to use the tools (G1, G3). The students who did use the tools did so in all phases of the project for different purposes. At the beginning of the project the tools were used to gain a better shared understanding of the situation (G5, G7, G8, G9, G10). For example, the tool was used by incorporating the affordances in a mind map. Although attention to the tools was lost in some groups in the middle of the project (G6, G8, G10), they were applied by other groups at this stage during the generation and validation of concept solutions (G5, G7). The tools contributed to an efficient design process because there was focus during concept development (G5). As G5 quoted: "The tools gave a lot of focus, especially during concept development". The tools were also applied at the end of the project, by analyzing the test insights of their prototype gained afterwards (G6, G7, G8, G10). The tools facilitated G10 agreement on how choices made regarding design solutions can be justified. When asked, the two partners, Roetz Bikes (P1) and X-Bike (P2) both indicated that the tools helped to (efficiently) arrive at desired (effective) solutions in a short period of time. P1 quoted: "I do think it just contributes to completeness of the solution and indeed, that the model offers the efficiency to reach agreement in a short time".

Despite the fact that two of the ten groups did not use the tools, we see that the tools are used in all phases in a variety of ways: from gaining shared understanding at the start of the project to generating solutions in the middle and evaluating solutions at the end of the project. Furthermore, both partners explicitly state that the tools have contributed to an efficient design process. From this we can conclude that, although improvements are needed, the tool as currently offered contributes to an efficient design process.

*3.3. Process Quality*

To assess the PQ of the applied design tools, all design students and partners were asked to name specific affordances (the sixteen affordances as presented in Section 2.1), mechanisms, and cards, and to name social consequences or effects for behavioral change of their work.

Six out of ten student groups (G2, G6 to G10) knew how to name specific affordances behind the cards applied during the project and how to relate them to choices made. During the interview, for example, G9 manages to articulate the affordance 'Emblems', which is about signaling and expressing information about the user's identity, as a possibility for self-expression of an end-user. The group justifies the designed intervention of self-made (customizable) stickers with the affordance 'Emblems' and describes the ability of 'Emblems' to increase the sense of ownership over bicycles. This substantiation is also reflected in the written rationale provided.

However, we also found evidence of weak internal validity of the tool. During the project G3 conducted an expert-interview with an external behavioral scientist during the process. From this expert-interview, an explanation emerged on how the subsequently chosen design solutions work, namely the Broken Window Theory (The Broken Window Theory provides a metaphor for disorder in residential areas. The theory links disorder and incivility within a community to later crimes, which in turn can be linked to inappropriate behavior. Conversely, the participants of this study reasoned that an orderly, tidy environment can ensure desirable, tidy behaviour.) [35]. This theory was mentioned by this group during the interview, was used in the report and during the final presentation of the work. During an interview with another group (G5) this theory is confirmed: one group member mentions the Broken Window Theory as a theory that has helped a lot in arriving at a good concept solution, while the Broken Window Theory is nowhere to be found in the report of G5. Without this theory being mentioned beforehand by the interviewer, P2 suggested the same Broken Window Theory as an explanation for the first intervention (the colored clustering of bicycles). We conclude that during the project different groups and partners substantiate some design choices outside the given theoretical framework of psychological ownership. This is an indication that the internal validity of the tool may be weak.

We identified that the students generally lost sight of the intended social consequences or effects for behavioral change of the interventions. All groups were asked to name the intended consequences as an effect of design solutions. None of the groups spontaneously mentioned the initially intended effects. After further questioning, in most cases the degree of felt ownership of the bicycles among users was first mentioned. Four groups (G4 to G7) looked for their answer to name consequences as an effect of their design solutions in the degree of user satisfaction. When asked, both partners spontaneously mentioned the intended effects of the design interventions. We can therefore conclude that the partners have not lost sight of the intended end result of the project. The partner of Roetz Bikes, responsible for repair work, spontaneously makes a link between the Bike Hospital intervention and closing the resource loop activities. He states that the turnover rate of the repairs (and therefore also products and materials) is the result of how quickly you are notified of defective bicycles. A Bike Hospital communicates the defective bikes directly and thus the resource flow is closed earlier. The partner of X-Bike, responsible for the app, mentions the effect that end users will be more careful in the use of equipment (bicycles) and that they will be more willing to help with certain [closing the loop] processes.

Moreover, we identified some affordances to take precedence over others. In the process, four groups (G4 to G7) discovered that a service that meets basic needs is a prerequisite for achieving higher psychological ownership. As G4 put it: " . . . the target group was actually dissatisfied with how the bicycle service is now being offered. There were already quite a few frustrations, so a logical choice for us was to remove these frustrations". From the exploratory field research they performed, the students found that many users of the service are dissatisfied, especially with the slowness of the app and a long wait before being able to use the bicycle as a result. Two groups (G4, G5) referred to this as

the affordances of Simplification and Enabling (the unlocking process with the app isn't easy and it doesn't enable the user). G7 went even further and took customer dissatisfaction as the starting point for a new concept of protest, which is actually a hyperbola for an end-user making a complaint. The group tested the extent to which dissatisfied bike users were willing to protest and found that no one was willing to do so. They referred to this as a lack of self-investment and attribute this to a lack of psychological ownership within TSH bicycle service. From this we can conclude that the affordances Simplification and Enabling may take precedence over other affordances to achieve the intended effect of, in this case, more involved end-users in repair & maintenance activities, a longer cycle life within a PSS. When asked, both partners noted that monitoring the end-user's expectations generated by the interventions is very important for the degree of perceived ownership and careful, responsible handling of the service. Involved users only exist with good service. This is in line with previous findings of groups G4, G5, G6 and G7.

In conclusion, a design tool can support designers by arriving at a conscious design process. Most student groups were able to identify the mechanisms initiated during the design process. However, the fact that none of the surveyed student groups could spontaneously name the initial effects of the project is an indication that the tool offered might be too focused on the mechanism rather that the intended effects, and therefore might not provide this support sufficiently.

### 3.4. Design Quality

In order to assess the DQ of the applied design tools, the bicycle repairer and nine end-users (U1 to U9) were asked about the consequences for behavioral change six months after the implementation, i.e., we looked for evidence of end-users involved in closing the loop activities, resulting in a longer lifespan. In addition, the repairperson was asked about the P.O. routes of self-investment (observing expressions of investment, such as making an effort and taking time) and intimate knowledge (observing or experience end-users prefer or appropriate certain exemplars).

When asked about the effects of the interventions (clustering bikes and bike check & bike hospital), the bicycle repairer indicated that users seem more involved in repair & maintenance activities and handle the bicycles more carefully. According to him, this is apparent (1) from a fewer number of defective bicycles and (2) from visibly different behavior of end-users. Most defective bikes are placed in the Bike Hospital as requested. "The difference with last year [before implementation of the intervention], also in terms of the number of broken bicycles, is quite big", the bicycle repairer concludes. For example, the bike repairer says, "[before the intervention] . . . we had a lot more bikes that were just thrown into a corner. So, they weren't put back in the right place at all. . . . That's a lot less now".

Asked about the P.O. in the route of self-investment, the bicycle repairer rarely observed that end-users check the bicycle extensively before putting the bicycle back. This picture is very much in line with the data from the interviews with end-users. These show that end-users do check the bicycles for defects and indicate this after use, but that the real check for defects is often already done during use of the bicycle and not afterwards. The bicycle is reported as defective if something has really been noticed during use or if there is a feeling that something is really wrong: "If the bicycle makes a crazy noise while cycling, I will report it" (U4). U5 indicates not to check the bike if there are no strange noises or difficulties.

Eight out of nine end-users do not experience performing the bicycle check as an investment of their own. Only one respondent (U9) said: "Yeah, it feels like a personal investment because I don't think everyone is doing it". When returning bicycles, the bicycle repairman sometimes notices irritation: "Yes, you see that sometimes, that they are just a bit irritated. But I'm just trying to be friendly". This irritation is not confirmed from the interviews with end-users. None of the end-users indicate that they have feelings of irritation when putting the bike back, performing a check or placing it in the Bike Hospital.

As U9 puts it: "It's normal for me to do. It feels like my responsibility as someone from the TSH community". The only annoyance observed by end-users is related to the Wi-Fi connection needed to open and close the bike lock. U5 states: " . . . the biggest investment is in the connection with your phone. Sometimes it is very annoying that the app reports that a number of bicycles are not available after you enter the number". Asked about the importance of the bike check, U9 answers: "Yes, I believe it is important. We are allowed to use the bicycles, so I think we should also take care of these bicycles".

Regarding the psychological ownership route of intimate knowledge, the bicycle repairman occasionally sees end-users looking for a specific bicycle. It sometimes happens that a student: " . . . stands at the blue wall and then you see him look like: oh, I have a preference for a certain number, but it's gone. Yes, then another student took him. Which is possible. And then they sometimes ask: hey, that's my number, right?" This is in line with the findings with the end-users. When asked, most end-users answer that they have a preference for a particular bicycle. All but one respondent indicated that they should take their favorite bicycle into account when reserving a bicycle. Most end-users can even spontaneously give the number of their favorite bike during the interview ("2241 is THE bike!"). The reasons for a particular exemplar were speed, speed/connection to the online service, saddle height and pedals.

In addition, the bicycle repairer also spontaneously described effects for his own work. He indicates that the interventions help him a lot in his work as a bicycle repairman and service provider. Clustering the bicycles and placing defective bicycles in the Bike Hospital gives him a good overview: "there is no longer one big mountain of bicycles". An overview of defective bicycles per colored cluster helps him in a good insightful workflow. He also describes that after the interventions were implemented there is sometimes a feeling of 'cooperation' between him and the end-user. The bicycle repairman is sometimes asked for help (for example when the app crashes or when reconfiguring the bicycle for use, such as adjusting the saddle height) and vice versa the repairman is sometimes helped: "Very occasionally someone comes to me please: dude, there is a problem with my bike, I put it in the hospital. Can you help me with the app? And then I help with that". This kind of reciprocity is reflected in U8's response, who states: "[the Bike repairer] is alone and I don't expect him to fix every bike within a day. So, if I can help with that, I don't mind". U9 describes the importance of the community as a reason to do what is asked in the intervention (do the bike check and put the bike back): "Communities like this need this kind of behavior, otherwise it is not a community and it won't work".

## 4. Discussion

The aim of this study was to arrive at answers to the main question: whether and how can designers be supported with a design tool, based on psychological ownership, to involve users in closing the loop activities? In order to give answers, we translated an analytical framework into a design tool and analyzed the design performance during a case study with TSH, design students, project partners, service providers and end-users. This study provided useful, both Top-Down and Bottom-Up insights for designing PSS with products with end-users being involved in closing resource loop, such as repair & maintenance activities.

### 4.1. Top-Down Insights

From our analysis from the Top-Down approach, new intermediate knowledge about the psychological ownership model and ensuing tools has been gained.

First, concerning the Process Efficiency, the results showed that the tool was perceived efficient in all stages in the design process. Using the tool for shared vocabularies and for project scoping, proved most beneficial. But, in contrast, there were some issues with the level of complexity and extensiveness of the tool. In addition, as G1 points out, " . . . more time is needed to get to grips with the cards". To improve Process Quality performance, we suggest that the tool will be supplemented, for example, in the form of a workshop.

Second, concerning the Process Quality, the results showed that most student teams kept sight of specific affordances identified at the start of the process, but had lost sight of the effects on object handling, responsible use and/or product lifetime. This indicates that the tools may be too focused on the mechanisms (the affordances) and too little on their effect as the end result. In order to improve PQ performance, we propose that in a next iteration the tool offers designers space to explicitly describe the intended effect as the end result and to reflect on this during the entire design process.

Third, an insight from the Top-Down approach is that one of the implemented interventions can be explained by a rival theory. This has implications not only for the design intervention, but also for the use and evaluation of design tools. As reflective design researchers, we not only want to know whether and *to what extent* design solutions are effective for the intentional result, but we also want to know *how* design solutions are effective for the intentional result. We want to know which mechanisms have been set in motion, in order to get a richer indication of the design solution in the context of the relevant practice. That is why we propose to see Design Quality as Design Validity + Design Effectiveness. Design Validity addresses the question of how the design solutions are effective for the intended result. Design Effectiveness hereby addresses the question of the extent to which the design solutions are effective for the intended result. Both Design Validity and Design Effectiveness are indicative of connecting design solutions in the context of the relevant practice, as described by Tromp & Hekkert [13]. To measure Design Validity, it seems advisable to further investigate the internal validity of the intervention, for example by means of an expert review.

*4.2. Bottom-Up Insights*

From our "Bottom-up" analysis, new intermediate knowledge about the psychological ownership model and ensuing tools has been gained.

First, our view on the potential effect of psychological ownership shifted while conducting our study. At the start of the study, our product-oriented scope was mainly to allow end users to handle the bicycles with care, with the direct consequence of extending the lifespan. However, while conducting this study, we came to the conclusion that the scope of psychological ownership is broader: through an experience of ownership, end users of a PSS can be involved in repair & maintenance activities, such as putting healthy and defective bicycles back in the right place. In this way, end users who experience ownership can be involved in closing the resource loop within a PSS.

Second, several student projects show that there is a lack of perceived user-friendliness of TSH bicycle service. This dissatisfaction of end-users might be linked to the affordances "Enabling" and "Simplification". Both the students and the partners stated that these affordances must actually be sufficient before other affordances can have effect. Although more evidence is needed, this might indicate a possible hierarchy in the affordances, with a prominent place for the Enabling and Simplification affordances (as shown in bold and cursive in Figure 7).

Third, the study showed that making a complaint or providing feedback can be seen as a self-investment of end-users. As a result, we can consider the ability for an end-user to file a complaint or provide feedback to the service provider as an additional affordance within the self-investment route (as shown as affordance '6. Giving feedback' in Figure 7).

Fourth, the role of and follow-up from the service provider plays a major role in the Bottom-Up findings. After all, it is this service provider who can show through the affordances of Enabling and Simplification that user effort (such as providing feedback on a slow-functioning loan app) is followed up. The role of the service provider is also reflected in the results from the interviews with TSH Bike repairer and the end-users. The implemented interventions, for example, appear to have unforeseen value for the performance of the work for the service provider (in this case the bicycle repairman). A degree of reciprocity has been observed between the bicycle repairer and end-users, with service provider and service user sometimes literally working together on repair

and maintenance activities to create a better and more sustainable service, which closes resource loops. This is in line with the study by Ackermann et al. [36], who regard repair & maintenance activities as a performance of both the consumer and the service provider [36]. The psychological ownership tool under study in its current form pays little or no attention to the role of the provider and the interplay between users and provider. It therefore seems advisable to include this role in the tool.

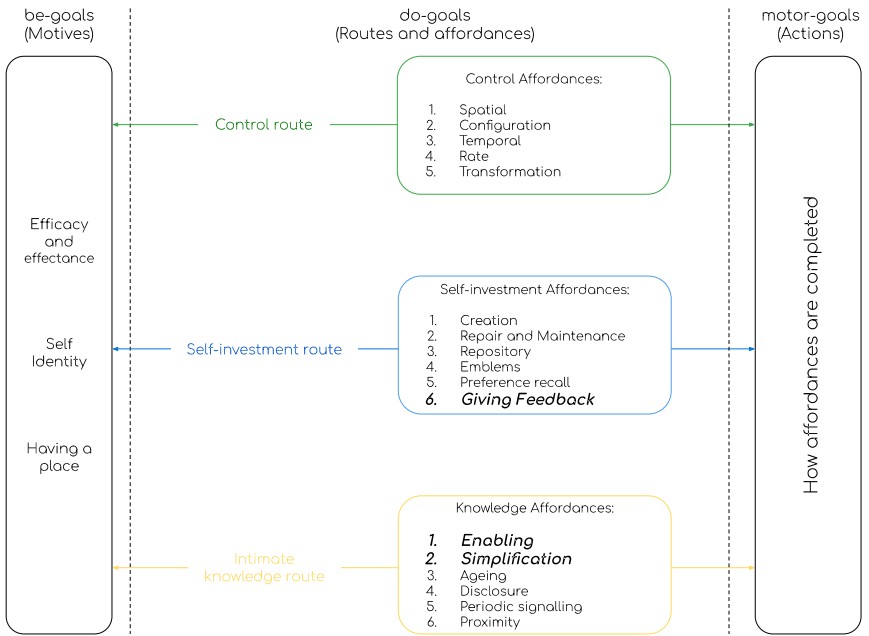

**Figure 7.** Revised psychological ownership Model.

It was decided to conduct a case study, which makes intensive research possible. However, like any study, this study also has limitations. Choosing a case study as a method forces one to make choices, which has consequences. For example, TSH's bicycle service is a PSS in one specific context, which concerns only products (bicycles) without energy consumption and which are mainly used intensively in certain countries. Following Boeije & Bleijenbergh [34], it is therefore advisable to conduct extensive follow-up research in which PSS can be investigated in multiple contexts (such as, for example, the use of washing machines or power tools).

## 5. Conclusions

Design researchers can play an important role in supporting designers and providers who deal with PSS in their daily practice by building up so-called intermediate knowledge. A design tool is an example of intermediate knowledge. In this study we showed how this knowledge about a design tool can be built up by acquiring Top-Down insights from the design process and literature on the one hand, and by acquiring Bottom-Up insights from the observed effects on the other. This research gives an indication that the application of psychological ownership affordances by designers can provide handles to achieve a greater involvement of end users in closing the resource loop activities within a PSS, such as repair & maintenance.

In summary, we can conclude that during the execution of this study, involving end users and the interaction between end users and service provider in repair & maintenance activities—and thereby contributing to closing the resource loop within this PSS—ultimately played an important role in the concept solutions created. We can also conclude that the developed model is utilitarian: the use of psychological ownership can play a role in involving both end users and service providers in closing the loop activities. This was not

self-evident before the start of the study. But that it has done so has provided new insights to support designers in designing to minimize environmental impact while using PSSs.

**Supplementary Materials:** The full version of the tools used during this study can be found online at https://www.hu.nl/onderzoek/projecten/ontwerpen-voor-duurzaam-gebruik-van-producten-binnen-product-dienst-systemen (accessed on 1 May 2022).

**Author Contributions:** Conceptualization, D.P.v.A., L.K. and R.v.d.L.; methodology, D.P.v.A., L.K., R.v.d.L. and B.E.; validation, D.P.v.A. and L.K.; formal analysis, D.P.v.A. and L.K.; D.P.v.A. and L.K.; resources, D.P.v.A.; data curation, D.P.v.A.; writing—original draft preparation, D.P.v.A.; writing—review and editing, L.K., R.v.d.L. and B.E.; visualization, D.P.v.A.; supervision, B.E.; project administration, D.P.v.A. All authors have read and agreed to the published version of the manuscript.

**Funding:** This research was funded by the Dutch Research Council, with Funding number KIEM.CIE.04.011. Application Date: 27 May 2019. Decision date: 7 January 2019.

**Institutional Review Board Statement:** Not applicable.

**Informed Consent Statement:** Informed consent was obtained from all subjects involved in the study.

**Data Availability Statement:** The data presented in this study are available on request from the corresponding author.

**Conflicts of Interest:** The funders had no role in the design of the study; in the collection, analyses, or interpretation of data; in the writing of the manuscript, or in the decision to publish the results.

## Appendix A. Protocol Interview Design Students

Questions 1–5: Process Quality [PQ] Does the designer show understanding and thoughtful considerations of the behavioral and social consequence of his/her design decisions throughout the process?

1.  Did you use the card set?

    a.  Yes: continue with question 2
    b.  No: continue with question 9

2.  Can you name specific cards?
3.  Can you name a specific moment in the design process? (Understand/Create/Deliver)
4.  Can you name specific design choices?
5.  Can you name the consequences of using the model/cards?

    a.  social consequences or consequences for behavioral change?

Questions 6–10: Process Efficiency [PE] Does the designer work efficiently, i.e., without too many detours?

6.  Have the cards helped you work more effectively?
7.  How have the cards helped you work more effectively? Can you cite an example?
8.  Could you have come up with a suitable design solution faster with or without a card set?
9.  For what reasons did you not use the cards?
10. What needs to happen that you are going to use the cards?

## Appendix B. Protocol Interview Bike Repairer TSH

Introduction: short explanation and purpose of my research (reducing environmental impact)

Questions about repairer's image before interventions (period up to and including March 2020)

1.  How did students (long-stayers) use the bicycles?
2.  How did you see this behaviour?

    - *Careful handling: think of checking for defects, repairing (tyre pressure, checking lights)*

- *Careless handling: cycling on pavements, do not climb the stairs over the tire gutter, otherwise.*

3. Did you experience ownership of the bicycles with students?
   - *Do you feel that users use the bicycles as they use their own bicycles?*

4. How did you see this back?
   - *Think, for example, of repairing bicycles or own baskets, stickers or lights on the bicycles, users who claim bicycles*

5. Has anything been done to encourage users to handle bicycles with care?
   - *Repair kit in garage, wall warnings, fines*

   Explanation of the interventions (without mechanisms of Psychological Ownership)

6. Do you see these interventions again and, if so, where and how? (not the behaviour of users, but the interventions themselves) Questions about Bike repairer's image after the interventions (period after March 2020)

7. How do students (long-stayers) use their bicycles now?

8. How do you see this?

9. Do you experience more ownership of the bicycles among students after the interventions?

   Context 1: Make a bike reservation

10. When users reserve a bike . . .

    . . . do users have a preference for a specific (copy of) bicycle? Why do you think? Do users take this into account when booking a sea bike? Do they feel more connected to a particular instance? [INTIMATE KNOWLEDGE]

    Context 2: Take a bike

11. When users take a bike . . .

    . . . do they know which cluster of bicycles they should be at? How do you see that?
    . . . do they have to adjust it often? How often? Do you estimate more or less often than before the interventions? What needs to be adjusted? (saddle) Do you see that users add parts (such as baskets) themselves. On the bike?
    . . . do they feel supported in this by The Student Hotel? (app/ bike/bike storage)
    . . . are they aware of the (new) text on the bicycle? What does this text do to them?
    Context 3: Use of the bike/destination

12. If users use a bicycle . . .

    . . . what is the purpose of that (usually)?
    How often are the bicycles used? (more or less in this period?)
    Do current bicycles help with that? (How) can that be improved?
    Context 4: Reset bike

13. When users return the bike . . .

    . . . do they check the bike then? What do they check for? How much time does that take them? What is their emotion about this? (irritation, discomfort, pride, concerned) Do users see this as a self-investment?

14. If there are any defects on the bicycle . . .

    . . . do users indicate this in the app? How much time does that take them? What is their emotion about this? (irritation, discomfort, pride, concerned)
    . . . do users know what to do with the bicycle? Do they know why? Find. Is this important?
    Context 5: Bike in repair

15. When a bike is under repair . . .

. . . do users know what happens to the bicycle? Do they know why? Do they know to what extent this contributes to the degree of environmental impact/circularity of the service? Do they think this is important? Do they think it is important to play a role in this?

16. Do the interventions help you in your work?
17. Do you think your role in the circularity of this service is important?
18. What do you think could be alternative explanations for changes in user behaviour?

**Appendix C. Protocol Interview Bike End-Users TSH**

Introduction

Who are you? Can you introduce yourself briefly?

Context 1: Before you can use a bike, you need to make a bike reservation

When you made a bike reservation . . .

. . . do you know which cluster this bike is part of? How do you know this?

. . . do you prefer a specimen or numbered bike? Why? Do you take this into account when you make a reservation? Do you feel attached to this specimen?

Context 2: Then you have to take a bike

When you take a bike . . .

. . . do you know at which cluster of bikes you have to be? Does this clustering help you? Does this feel as 'your' cluster?

. . . do you often have to reconfigure this bike before you can use it? Where and how (i.e., saddle height)?

Do you, in some way, make the bike yours? (putting your own lights or a basket on it)

Context 3: After taking the bike you use the bike for a certain goal.

When you use a bike . . .

. . . for what purpose or aim you use this bike?

Does the bike helps you to achieve this?

Context 4: After using the bike you have to put back the bike

When you put the bike back . . .

. . . do you check the bike? On which parts do you check? How much time does this take? How does this feel? (i.e., annoying because you are in a rush, inconvenient or proud, involved).

. . . Do you know why you have to check the bike? Is this important to you? Do you consider this as a personal investment in the service?

When the bike seems to have defects after usage . . .

. . . do you state this in the App? How? . . . do you know what to do with the bike after states this in the app?

. . . Do you know why you have to do this? Is this important to you?

. . . How does this feel? (i.e., annoying, inconvenient, ashamed or proud, involved?

Context 5: A Bike in the bike hospital is in repair

When the bike is in repair . . .

. . . do you know what actually happens with the bike? Do you know why? Do you know to what extent TSH bikes are circulair?

. . . Is this important to you? Is it important to you to play a role in this circularity?

During the service as a whole . . .

. . . are you aware of the QR code and text at the bike?

What does this text say to you? Do you feel addressed? How and why?

Do you feel responsible for the service being durable?

Do you know in what way you can form and/or reduce environmental impact while using the service?

Is this important to you?

(How) can this bike service be improved to be more meaningful for you?

Many thanks!

## Appendix D. Design Focus and Rationale by Student Group

| | Design Focus | Rationale (Written Rapport) | Rationale (Oral) |
|---|---|---|---|
| Student Group 1 (G1) HU Studio Gif | Bicycles per corridor with color, where one person is responsible per corridor. Adapting the bike to customer needs: bag rack and possibility to 'pimp'. | Affordances P.O. not mentioned in reporting. However, 'togetherness' and 'social control' to increase a sense of responsibility over shared items. | Affordances P.O. not mentioned. |
| Student Group 2 (G2) Hu happy design club | Bike check in app (check before use) and 'Holy Bike'. | Affordances P.O. not mentioned in reporting. | From checklist to visualization of the problem: simplifying, explaining and making reporting possible (Simplification, Repository). "In addition, disclosure really applies to the Holy Bike". |
| Student Group 3 (G3) HU Four minus one | Redesign bicycle shed with art and color + instagram. | Affordances P.O. not mentioned in reporting. Broken Window Theory as a rationale for redesign. | 'spatial' & 'preference recall'. No further research has been done on this. |
| Student Group 4 (G4) HU Overall | Take Care; Plasters and bike hospital, physical keys | Repair & Maintenance and Simplification | Repair & Maintenance and Simplification |
| Student Group 5 (G5) HU Kamer-plantjes | Bike Postive: color, numbers, statistics, quotes, infirmary | Affordances P.O. not mentioned in reporting. Tali Sharot's 'Sense of Control' theory and social proof as a rationale. | Affordances P.O. not mentioned. |
| Student Group 6 (G6) HU LMTD | Bike Tour and Cycling course + cycling per course and points system. | Affordances P.O. not mentioned in reporting. However, investing is mentioned as a rationale for participating in the bike course. | Looked at Emblems. So that's the self-investment route. And simplification was very important and enabling for us, so I think more people have worked towards that intimate knowledge route than self investment too. |
| Student Group 7 (G7) TU/e field | Avatar in app on bike | Self-investment route: familiar bike settings, having a depiction of themselves on the bike (avatar), having a personalized avatar through creation. | Self-investment and creation, that was kind of the route and before that it went like this: We want self investment, that includes creation and emblems and later specifically creation. |

| | Design Focus | Rationale (Written Rapport) | Rationale (Oral) |
|---|---|---|---|
| Student Group 8 (G8) TU/e Showroom | Different types of bicycles | We suggest that the Baxter model has a pyramid shaped hierarch relating to psychological ownership of shared services. In this hierarchy, identifying meaning in non-ownership stands on the basis. Without meaning in non-ownership over a shared service, it provides no added value over actually owning the product. In many shared ownership services, the value in non-ownership is immediately clear. For example, the service can be cheaper than owning the product, more sustainable, or more conveniently accessible at multiple places. | With protest, of course, we really started to focus on that self-investment route. So I think that was also an important step, especially looking back to the baxter model, that we really made the choice there: okay, we're going to look at that route now, and focus specifically on that. |
| Student Group 9 (G9) TU/e Overflow | Self-made stickers on bike for each use | Emblems, Configuration, Temporal and Preference Recall. | The emblems affordance and Contamination. |
| Student Group 10 (G10) TU/e Lab | Physical personalized element in frame | Personalisation | Transformation, creation en emblems. I find those specific affordances partly present in a certain way. |

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
