# Peer review of "A Psychological Ownership Based Design Tool to Close the Resource Loop in Product Service Systems: A Bike Sharing Case"

_sustainability, doi:10.3390/su14106207_

Round 1

Reviewer 1 Report

No significant comments.
The objectives of the article are documented in the substantive part.
Model visualizations affect the cognitive value of the article.
For discussion, it can be concluded in conclusion whether the developed model is utilitarian? pp.15

Author Response

Dear reviewers,

First of all, many thanks for your extensive and constructive reviews. It helps a lot to improve the research paper. With this letter we answer to all your comments and suggestions.

Regarding the initial comment on data saturation in our survey, we've included the following text in the modified paper:

'To arrive at saturation of answers to the main question, all interviews with design students and partners were transcribed and data was coded. Because the existing theoretical framework of Design Performance was the starting point for this research, it was decided to code deductively with indicators [36]. The concepts Process Efficiency, Process Quality and Design Quality have been unraveled by one researcher into concrete observable indicators. The proposed indicators were submitted to a second researcher as a check and adjusted in consultation. The text fragments from the transcripts are then linked to the indicators and compared with each other. From this comparison, the research team has arrived at saturated answers to the main questions: when and how the team members applied the design tools, how these obstructed or supported the design process, if the students and partners showed shared understanding of the intended consequences of their actions and whether and to what extent design solutions are effective for the intentional result in the eyes of the bicycle repairer and end-users.'

With this we not only show the data analyzing process as described by Boeije & Bleijenbergh (2019), but we also indicate that the saturation in answers does not only exist with the 42 design students but also, additionally, with the project partners.

Regarding the second comment on 'expanding the literature by including more recent studies', we have supplemented the introduction with four studies on psychological ownership from 2021 and 2022. These recent studies confirm our findings on the positive influence of psychological ownership on stewardship, but confirm also that there is still a lack of knowledge about the application of these insights in the design process of use without ownership.

Then the reviewers suggested considering the following.

  1. The practical usability/usefulness of the model.

In response to this suggestion, we have supplemented the conclusion with the following text:

'We can also conclude that the developed model is utilitarian: the use of psychological ownership can play a role in involving both end users and service providers in closing the loop activities.'

  1. Research limitations and recommendations for further research.

In response to this suggestion, we have supplemented the discussion with the following text:

'It was decided to conduct a case study, which makes intensive research possible. However, like any study, this study also has limitations. Choosing a case study as a method forces one to make choices, which has consequences. For example, TSH's bicycle service is a PSS in one specific context, which concerns only products (bicycles) without energy consumption and which are mainly used intensively in certain countries. Following Boeije & Bleijenbergh [36], it is therefore advisable to conduct extensive follow-up research in which PSS can be investigated in multiple contexts (such as, for example, the use of washing machines or power tools).'

  1. The use of figures.

We like to thank reviewer 3 for this suggestion. Nevertheless, we follow reviewer 1 (“Model visualizations affect the cognitive value of the article”) in our belief that all numbers have their value and would like to keep them in the paper.

  1. Better argumentation of why the existing model needed to be improved.

Following this suggestion, we have added the following text on page 4:

'One of the most important conditions for using a design tool as a method is that the user has confidence in the tool. As Daalhuizen [33] puts it: “Trust in a method reflects both confidence in one’s ability to use the method in a way that yields desirable results as well as confidence in the applicability of the method itself to a certain goal-domain”. We do not believe that simply offering a tool in a research paper textually gives this confidence to the users. In order to maximize this trust among users, it was decided to make adjustments and additions that translate the original framework to a design tool to make it more accessible and applicable for designers (as shown in figure 2).'

We look forward to your response with great confidence. If you have any questions, please contact Dirk Ploos van Amstel.

Yours sincerely,

Dirk Ploos van Amstel

Lenneke Kuijer

Remko van der Lugt

Berry Eggen

Reviewer 2 Report

The aim of this study is to develop and evaluate a design tool in the context of product service system (PSS), based on psychological ownership, though a case study that focus on a bicycle sharing service. This paper is well-written and has clear contributions. There are some suggestions that can enhance the quality of this study:

  1. It would be useful to to highlight the research limitations and consequently future work.
  2.    To increase the clarity, the authors are advised to allocate a section to highlight the practical implications of this work. 
  3. It would be beneficial to enrich  the introduction section by discussing related and recent literature.  

Author Response

(The authors gave the same response as above.)

Reviewer 3 Report

The topic is intriguing and actual. I would not consider figure 1 in the introductory part, but in the following section. A better argumentation on the necessity to increase the ownership model could be added. The originality of the modified PO model could be upgraded. The paper presents too m any figures, without extensive description of the logical connections among them and the utilised sequence (why there is a figure no. x and then x+1). The study seems extensive, although the number of subjects is not supportive for the relevance of the results (e.g. 42 students). The location of the study Amsterdam is suitable for the topic of the investigation, but can bring some uncertainty on the conclusions, as not everywhere the use of bikes is so common. 

Author Response

(The authors gave the same response as above.)

Round 2

Reviewer 3 Report

I think the paper is publishable in this updated version, after English proof.